# Oral Antiplatelet Therapy for Secondary Prevention of Non-Cardioembolic Ischemic Cerebrovascular Events

**DOI:** 10.3390/jcm10081721

**Published:** 2021-04-16

**Authors:** Leonardo De Luca, Elisa Bellettini, Dario Di Maio, Enrico Natale, Rita Lucia Putini, Sabrina Anticoli, Furio Colivicchi, Paolo Calabrò, Francesco Musumeci, Domenico Gabrielli

**Affiliations:** 1Division of Cardiology, Department of Cardiosciences, Azienda Ospedaliera San Camillo-Forlanini, 00152 Rome, Italy; ebellettini@scamilloforlanini.rm.it (E.B.); enatale@scamilloforlanini.rm.it (E.N.); rlputini@scamilloforlanini.rm.it (R.L.P.); dgabrielli@scamilloforlanini.rm.it (D.G.); 2Division of Cardiology, Department of Translational Medicine, University of Campania Luigi Vanvitelli, 81100 Caserta, Italy; darioddm90@gmail.com (D.D.M.); paolo.calabro@unicampania.it (P.C.); 3Stroke Unit, Azienda Ospedaliera San Camillo-Forlanini, 00152 Rome, Italy; santicoli@scamilloforlanini.rm.it; 4Division of Cardiology, S. Filippo Neri Hospital, 00135 Rome, Italy; furio.colivicchi@gmail.com; 5Cardiac Surgery Unit and Heart Transplantation Center, Department of Cardiosciences, Azienda Ospedaliera San Camillo-Forlanini, 00152 Rome, Italy; fr.musumeci@gmail.com

**Keywords:** antiplatelet therapy, aspirin, clopidogrel, ticagrelor, stroke, TIA

## Abstract

Stroke is the leading cause of disability and mortality worldwide. After an acute cerebrovascular ischemia, recurrent vascular events, including recurrent stroke or transient ischemic accidents (TIA), occur in around 20% of cases within the first 3 months. In order to minimize this percentage, antiplatelet therapy may play a key role in the management of non-cardioembolic cerebrovascular events. This review will focus on the current evidence of antiplatelet therapies most commonly discussed in practice guidelines and used in clinical practice for the treatment of stroke/TIA complications. The antiplatelet therapies most commonly used and discussed are as follows: aspirin, clopidogrel, and ticagrelor.

## 1. Introduction

Cerebrovascular events are a common cause of mortality and disability worldwide [1]. Due to early identification of symptoms as well as the improvement of primary prevention strategies and in-hospital management, the incidence of stroke and the related short-term and long-term mortality has been steadily decreasing in recent decades, especially in subjects aged over 65 years and in the most advanced countries [2]. The goal for the management of patients with ischemic stroke (IS) and transient ischemic attacks (TIA) is to decrease the early risk of recurrent cerebrovascular events and long-term incidence of severe nonfatal and fatal vascular events [1,3,4]. Notably, a recurrent stroke has been estimated in a range of 10% to 20% within the first 3 months after the index event, with most recurrent strokes occurring within the first 2 days [5,6]. For this reason, international guidelines currently recommend early medical assessment [7]. In this regard, antithrombotic therapy is a cornerstone for secondary prevention of cerebrovascular accidents [7]. Although there is consensus about the management of IS of cardioembolic origin, data regarding the correct antithrombotic therapy for secondary prevention of non-cardioembolic cerebrovascular events are conflicting.

In this paper, we focus on ischemic cerebrovascular events of non-cardioembolic origin, reviewing different antithrombotic strategies aimed to prevent ischemic recurrences in this high-risk population.

## 2. Secondary Prevention: Antiplatelet Therapy

International guidelines recommend starting antiplatelet treatment with acetylsalicylic acid (ASA) (≤325 mg o.d.) within 24–48 h from symptoms onset after non-cardioembolic IS [7]. The rationale of antiplatelet therapy is to limit thrombosis on ulcerated atherosclerotic plaques and subsequent distal embolization.

Despite the large use of ASA, an important number of recurrent events persist, ranging from 10% to 15% in the first 90 days [8]. For this reason, different and more intensive antiplatelet regimens were tested. In Table 1, we resumed the main characteristics and differences in the study design of the four main trials on antiplatelet agents tested in this field. Table 2 shows the comparison in efficacy and safety end points of these trials.

## 3. Low Dose ASA

ASA is an irreversible inhibitor of cyclooxygenase, the major enzyme in the synthesis of prostaglandin and thromboxane, a potent platelet activator. In 1994, an analysis of trials on long-term antiplatelet therapy in patients with previous myocardial infarction (MI), stroke, or TIA demonstrated that ASA avoids about 40 serious vascular events per 1000 patients treated for a few years [13]. In 1997, two randomized trials tested the efficacy of ASA in the acute phase of IS: the CAST (Chinese Acute Stroke Trial) [14] and the IST (International Stroke Trial) [15].

The CAST was entirely conducted in China and included 21,106 patients admitted for an acute IS [14]. Patients were randomly assigned to 160 mg/day of ASA or a placebo within 48 h from symptoms onset. ASA produced a significant 14% proportional reduction in mortality (3.3% vs. 3.9%; *p* = 0.04) and in recurrent IS (1.6% vs. 2.1%; *p* = 0.01), with a 12% proportional risk reduction in the composite in-hospital endpoint of death or non-fatal stroke at 4 weeks (5.3% vs. 5.9%; *p* = 0.03), corresponding to an absolute difference of 6.8 fewer cases per 1000. Hemorrhagic stroke was slightly but not significantly increased (1.1% vs. 0.9%) [14].

The IST was a randomized, open trial of up to 14 days of antithrombotic therapy started immediately after IS in 19,435 patients in Western countries. In a factorial design, half of the patients were allocated to unfractionated heparin (5000 IU or 12,500 IU twice daily), and half were allocated to “avoid heparin”; hence, half were allocated to ASA (300 mg daily), and half were allocated to “avoid aspirin”. Heparin (low and medium dose combined) did not significantly reduce deaths or non-fatal recurrent stroke at 14 days or 6 months. Conversely, among aspirin-allocated patients, there were non-significantly fewer deaths within 14 days (9.0% vs. 9.4%) and a non-significant trend towards a lower death rate or dependent at 6 months (62.2% vs. 63.5%). After adjustment for baseline prognosis, the benefit from aspirin was significant (*p* = 0.03) at 6 months. As in CAST, ASA did not produce a significant increase in hemorrhagic strokes (0.9% vs. 0.8%) [15].

Combining two trial populations, ASA, promptly started in the acute setting, showed a highly significant reduction in recurrent IS (1.6% vs. 2.3%; *p* < 0.000001). Safety was confirmed by an increase of only 2 per 1000 in hemorrhagic stroke or hemorrhagic transformation of the original infarct (1.0% vs. 0.8%; *p* = 0.07). Interestingly, subgroup-specific analyses did not reveal significant heterogeneity, leading to conclude ASA benefits do not substantially differ with respect to age, sex, level of consciousness, atrial fibrillation, computed tomographic (CT) findings, blood pressure, stroke subtype, or concomitant heparin use [16].

Subsequent metanalyses confirmed the benefits of ASA, with a dose ranging between 75 and 150 mg/day, for secondary prevention of patients with IS in terms of reduction in recurrence of cerebrovascular events and risk of serious vascular events [17,18]. Notably, the improvement in ischemic outcomes outnumbered the increased risk in bleedings, with a net risk-benefit ratio favoring ASA [17,18].

## 4. Clopidogrel

Clopidogrel selectively inhibits the adenosine diphosphate platelet P2Y12 receptor, blocking the glycoprotein (GP) IIb/IIIa complex, thereby inhibiting platelet aggregation. Clopidogrel, 75 mg/day, was firstly tested in comparison to 325 mg/day of ASA 325 mg/day in the CAPRIE (Clopidogrel Versus Aspirin in Patients at Risk of Ischemic Events) trial on a heterogeneous group of patients with prior MI, prior stroke, or peripheral arterial disease (PAD). It provided an additional 8.7% relative-risk reduction, with an overall safety profile at least as good as that of medium-dose ASA. Indeed, severe upper gastrointestinal discomfort, intracranial hemorrhage (ICH), and gastrointestinal hemorrhage were numerically more frequent with ASA, similar to non-fatal primary ICH or hemorrhagic death (0.39% vs. 0.53%). Among 59% of patients with previous stroke due to atherothrombotic disease, clopidogrel produced a slight relative-risk reduction of 7.3% (95% CI, −5.7 to 18.7; *p* = 0.26), even if the trial was not powered to perceive a concrete effect in subgroups. Considering patients with symptomatic atherosclerotic disease, prior MI, or prior stroke, the benefits of clopidogrel were amplified (ARR 1.4% at 1 year and 3.4% at 3 years; 95% CI, 0.2 to 7.0; RR 14.9%; 95% CI, 0.3 to 27.3; *p* = 0.045) [19].

The association of clopidogrel and ASA was investigated in two small population trials: CARESS (Clopidogrel and Aspirin for Reduction of Emboli in Symptomatic Carotid Stenosis) and CLAIR (Clopidogrel Plus Aspirin Versus Aspirin Alone for Reducing Embolization in Patients With Acute Symptomatic Cerebral or Carotid Artery Stenosis), by microembolic signals (MES) [20,21]. Indeed, as MES, detected by transcranial Doppler ultrasound, are a surrogate marker of future stroke and TIA risk, they were used to evaluate antiplatelet therapy efficacy in patients with symptomatic cerebral or carotid stenosis. In both trials, dual antiplatelet therapy (DAPT) with clopidogrel plus ASA was more effective than ASA alone in reducing microembolic signals, namely asymptomatic embolization [20,21]. Subsequently, the MATCH (Management of Atherothrombosis with Clopidogrel in High-risk patients) trial tested the efficacy of DAPT in terms of clinical outcomes in patients with recent IS or TIA (mean time to randomization of 26.5 days) and at least one additional vascular risk factor. The principal causes of stroke were small-vessel disease (SVD) (53%) and large-artery atherosclerosis (34%). DAPT produced a non-significant relative risk reduction of 6.4% (95% CI, −4.6 to 16.3) in the composite of IS, MI, vascular death, or rehospitalization for acute ischemia. In addition, bleeding events, including life-threatening bleedings, were almost doubled in the DAPT group [22]. Similar data emerged from the SPS3 (Secondary Prevention of Small Subcortical Strokes) trial, aimed to investigate the effectiveness of long-term DAPT for secondary prevention in magnetic resonance imaging confirmed lacunar infarcts. After 3.4 years of follow up, DAPT with clopidogrel (75 mg/day) plus ASA (325 mg/day) did not reduce the risk of recurrent stroke, neither ischemic stroke, disabling stroke, nor fatal stroke, with a concomitant increase in bleedings, including extracranial and gastrointestinal bleeding. It is noteworthy that all-cause mortality increased (HR 1.52; 95% CI, 1.14 to 2.04; *p* = 0.004) with similar trends for different causes of death [23].

The CHARISMA (Clopidogrel for High Atherothrombotic Risk and Ischemic Stabilization, Management, and Avoidance) trial investigated dual therapy in comparison with ASA alone in a population with multiple atherothrombotic risk factors or documented cardiovascular disease. However, in such a heterogeneous population, dual therapy failed to prove its efficacy. Furthermore, the risk of moderate-to-severe bleeding was increased [24]. A post hoc analysis on the subgroup of patients with documented vascular disease revealed a significantly lower rate in the composite of cardiovascular death, MI, IS, death from any cause (7.3% vs. 8.8%; HR 0.83; 95% CI, 0.72 to 0.96; *p* = 0.01), and hospitalizations for ischemia (11.4% vs. 13.2%; HR 0.86; 95% CI, 0.76 to 0.96; *p* = 0.008). No significant differences in the rate of severe bleeding were reported, albeit there was a significant increase in moderate bleeding (HR 1.60; 95% CI, 1.16 to 2.20; *p* = 0.004) [25]. However, a post hoc analysis focusing on patients with prior stroke showed a comparable rate of recurrence and functional severity of stroke at follow-up between DAPT vs. ASA alone (95% CI, −4% to 29%; *p* = 0.12). Finally, among patients with a qualifying diagnosis of TIA or IS, the rate of recurrent stroke was 5.4%, with a non-significant relative risk reduction of 20% by DAPT (95% CI, −3% to 38%) [26].

After the unconvincing and conflicting data on the role of long-term DAPT in secondary prevention of stroke, studies on the potential role of early antiplatelet therapy after an IS event were designed and conducted. The CHANCE (Clopidogrel in High-Risk Patients with Acute Nondisabling Cerebrovascular Events) trial enrolled 5170 Chinese patients affected by minor stroke or high-risk TIA [12]. Minor stroke was defined as a NIHSS (National Institutes of Health Stroke Scale) score of ≤3. High-risk TIA was defined as a moderate-to-high risk of stroke recurrence with an ABCD2 (Age, Blood Pressure, Clinical Features, Duration, Diabetes) score of ≥4. Patients were randomly assigned to receive clopidogrel at an initial dose of 300 mg, followed by 75 mg/day for 90 days, plus low-dose ASA (75 mg/day) for the first 3 weeks, or placebo plus ASA (75 mg/day for 90 days), within 24 h after symptoms onset. The combined therapy reported a decreased rate of stroke (ischemic or hemorrhagic) at 90 days (8.2% vs. 11.7%: HR 0.68; 95% CI, 0.57 to 0.81), fatal or disabling stroke (5.2% vs. 6.8%: HR 0.75; 95% CI, 0.60 to 0.94; *p* = 0.01), and IS (7.9% vs. 11.4%: HR 0.67; 95% CI, 0.56 to 0.81; *p* < 0.001). Interestingly, moderate or severe hemorrhage and hemorrhagic stroke did not increase (*p* = 0.73), nor did the rate of any bleeding event (2.3% vs. 1.6%: HR 1.41; 95% CI, 0.95 to 2.10; *p* = 0.09). Post hoc analysis also found a small but measurable reduction in poor functional outcome (95% CI, 0.03 to 3.42; *p* = 0.046) [11]. A subsequent report of 1-year outcomes showed a persistent absolute risk reduction of stroke (3.5% at 3 months and 3.4% at 1 year) after an initial period of greater advantage [12].

These data have been largely confirmed by the POINT (Platelet-Oriented Inhibition in New TIA and Minor Ischemic Stroke) trial that enrolled 4881 patients in Western countries within 12 h from a mild, non-cardioembolic, acute IS (NIHSS score ≤ 3) or a high-risk TIA (ABCD2 ≥ 4) [10]. As in the CHANCE trial, patients received clopidogrel 75 mg/day plus ASA (50 mg/day to 325 mg/day) or ASA alone. Primary efficacy outcome of major ischemic events (a composite of IS, MI, or death from an ischemic vascular event) was significantly reduced by DAPT (5.0% vs. 6.5%; HR 0.75; 95% CI, 0.59 to 0.95), mainly driven by a significant IS reduction (HR 0.72; 95% CI, 0.56 to 0.92). However, a significant increase in major bleeding was reported (HR 2.32; 95% CI, 1.10 to 4.87) without significant differences detected between groups in the rates of hemorrhagic stroke or symptomatic ICH. Notably, a relevant difference in time distribution of ischemic events and bleeding events was noted in the DAPT arm. Indeed, the incidence of recurrent ischemic events seemed to be enhanced in the first month, so that the benefit of DAPT was greater in the first 7 days and in the first 30 days than at 90 days (*p* = 0.04 for days 0 to 7, and *p* = 0.02 for days 0 to 30), whereas the risk of bleeding was greater from 8 to 90 days than during the first week from the index event (*p* = 0.04 for days 8 to 90, and *p* = 0.34 for days 0 to 7). These data could explain the difference in safety outcome registered in CHANCE where patients received a different dose of ASA (75 mg/day) added to clopidogrel, for a shorter period of time (21 days). The lower dose of ASA and the limited duration of DAPT could explain the reduced rate of bleeding events registered in the CHANCE trial compared to the POINT trial [10]. Nevertheless, data from the POINT trial supported those from the CHANCE trial, proving the role of DAPT in the acute setting of mild non-cardioembolic IS.

## 5. Ticagrelor

Ticagrelor is a potent, reversibly binding, direct-acting P2Y12 receptor antagonist. It proved its superiority over clopidogrel in patients with acute coronary syndrome reducing ischemic events, including death from vascular cause, MI, and stroke, with no significant increase in major bleedings [27,28,29,30]. The SOCRATES (Acute Stroke or Transient Ischemic Attack Treated With Aspirin or Ticagrelor and Patient Outcomes) trial firstly tested the efficacy and safety of ticagrelor (loading dose of 180 mg followed by 90 mg bid) in the contest of acute ischemic cerebrovascular events, in comparison with ASA (loading dose of 300 mg followed by 100 mg/day), enrolling 13,199 patients ≤ 24 h from non-severe acute IS or high-risk TIA. Study population was strictly selected: NIHSS score of 5 or lower in the case of stroke, ABCD2 score of 3 or more in the case of TIA, or symptomatic intracranial or extracranial arterial stenosis [9]. Patients were not eligible in cases of suspicious cardioembolic cause, if they underwent intravenous or intraarterial thrombolysis or mechanical thrombectomy, or if they needed any specific antiplatelet or anticoagulant therapy. All patients underwent CT or magnetic resonance imaging (MRI) scans in order to rule out intracranial bleeding or other conditions. At 90 days, the primary end point, a composite of IS, MI, or death, occurred in 6.7% of patients in the ticagrelor arm compared with 7.5% in the ASA arm (HR 0.89; 95% CI, 0.78 to 1.01; *p* = 0.07). Although all analyses of secondary endpoints were considered exploratory, it was not appropriate to make conclusions regarding significance. However, IS was reduced by ticagrelor (HR 0.87; 95% CI, 0.76 to 1.00; nominal *p* = 0.046). Interestingly, ticagrelor did not cause safety concerns and had no increase in any safety end point [8]. Among the 3081 patients (23% of the overall population enrolled) with a potentially symptomatic ipsilateral atherosclerotic stenosis, ticagrelor produced a relative risk reduction of 32% (6.7% vs. 9.6%; HR 0.68; 95% CI, 0.53 to 0.88) over ASA in the occurrence of the primary endpoint, without significant differences in the proportion of any bleeding events. The efficacy in this subtype of stroke is explained by the predominance of white thrombus (rich in platelet aggregates) in the mechanism of IS/TIA in large artery atherosclerosis [9]. A following analysis assessed the efficacy of ticagrelor in patients who received ASA before randomization. Since the antiplatelet effect of ASA persisted during the first week, patients later assigned to ticagrelor effectively received a DAPT for the first week after the acute event, the period at highest risk. Data on this group revealed a 24% relative risk reduction (HR 0.76; 95% CI, 0.61 to 0.95; *p* = 0.02) with ticagrelor, although no significant treatment by prior ASA interaction was found [31].

Subsequently, the THALES (The Acute Stroke or Transient Ischemic Attack Treated With Ticagrelor and ASA for Prevention of Stroke and Death) trial randomly assigned patients with high-risk TIA or mild to moderate acute non-cardioembolic IS who were not undergoing thrombolysis or thrombectomy, to combined ASA and ticagrelor, versus ASA alone [5]. Inclusion criteria were similar to the SOCRATES trial, with an ABCD2 score for TIA of 6 or more. Treatment was started within 24 h, with a loading dose of 180 mg followed by 90 mg twice daily for ticagrelor, and a loading dose of 300–325 mg followed by 75–100 mg/day for ASA. The DAPT regimen established its efficacy in reducing the rate of IS or death at 30 days by 17% (5.5% vs. 6.6%; HR 0.83; 95% CI, 0.71 to 0.96; *p* = 0.02), and IS by 21% (5.0% vs. 6.3%; HR 0.79; 95% CI, 0.68 to 0.93; *p* = 0.004). However, ticagrelor-ASA was associated with a higher risk of severe bleeding, as defined according to GUSTO criteria (HR 3.99; 95% CI, 1.74 to 9.14; *p* = 0.001). Even the composite of ICH or fatal bleeding was increased (0.4% vs. 0.1%; HR 3.66; 95% CI, 1.48 to 9.02; *p* = 0.005). It was estimated that the benefit from treatment with ticagrelor-ASA as compared to ASA alone would result in a number needed to treat of 92 to prevent one primary outcome event, and a number needed to harm of 263 for severe bleeding [5], opening a possible role for ticagrelor in secondary prevention after IS in patients with low bleeding and high ischemic risk. A prespecified analysis in patients with and without ipsilateral, potentially causal atherosclerotic stenosis ≥30% of cervical-cranial vasculature, showed a magnified efficacy of ticagrelor. The primary endpoint of stroke or death at 30 days occurred in 8.1% patients randomized to ticagrelor versus 10.9% in the placebo arm (*p* = 0.023), with an almost 30% reduction in the risk of stroke recurrence (*p* = 0.02). Once again, Kaplan–Meier curves revealed the efficacy was greater in the first period from index event, exactly within the first 10 days [32].

## 6. Conclusions

Current guidelines recommend ASA administration ≤48 h in case of non-cardioembolic acute IS [7]. Regarding the role of the DAPT regimen based on ASA plus clopidogrel in the acute setting, data from the CHANCE and POINT trials changed the level of recommendation, as the association of the two antiplatelet drugs demonstrated to produce a significant benefit in terms of reduction in ischemic risk without major bleeding concerns. Indeed, DAPT is now indicated for patients presenting with minor non-cardioembolic IS (NIHSS score ≤ 3) who did not receive thrombolytic therapy, and should be started ≤24 h from symptoms onset and continued for 21 days [7]. Conversely, data on the use of thrombolysis in patients on DAPT and the role of DAPT for long-term prevention are less conclusive. Although current guidelines allow for thrombolysis in patients with stroke under DAPT [7], doubts remain on the safety and the beneficial effects of thrombolytic therapy in this setting, leading to undertreatment. Regarding long-term DAPT, MATCH, SPS3, and CHARISMA, trials have not convincingly proved a net clinical benefit from clopidogrel–ASA association, due to a non-substantial prevention of recurrent events, worsened by a significant increase in major bleeding [7].

Lastly, guidelines did not include results from the THALES trail and consequently do not recommend adding ticagrelor to ASA. However, this trial attested an ischemic benefit from ticagrelor/ASA treatment compared to ASA alone among patients with a mild-to-moderate acute non-cardioembolic ischemic stroke or TIA. Notably, the improvement in stroke or death prevention seems to offset bleeding concerns. Therefore, ticagrelor may have a role in selected patients who present a particularly high thrombotic profile, and a low hemorrhagic risk, as those with atherosclerotic stroke.

Finally, in patients taking ASA at the time of the incident stroke, the benefit of switching to an alternative antiplatelet agent or combination therapy is not well established and needs further study [33].

## Figures and Tables

**Table 1 jcm-10-01721-t001:** Principles, trials, designs, and characteristics.

	SOCRATES [9]	THALES [5]	POINT [10]	CHANCE [11,12]
**Inclusion criteria**	Stroke: NIHSS score ≤ 5 TIA: ABCD2 score ≥ 4 Symptomatic intracranial or extracranial arterial stenosis CT or MRI scan rule to out ICH or other conditions	Stroke: NIHSS score ≤ 5 TIA: ABCD2 score ≥ 6 Symptomatic intracranial or extracranial arterial stenosis CT or MRI scan to rule out ICH or other conditions	Stroke: NIHSS score ≤ 3 TIA: ABCD2 score ≥ 4 CT or MRI scan to rule out ICH or other conditions	Stroke: NIHSS score ≤ 3 TIA: ABCD2 score ≥ 4 mRS ≤ 2
**Exclusion criteria**	Cardioembolic cause i.v or i.a. thrombolysis Mechanical thrombectomy Planned percutaneous revascularization Planned anticoagulation or antiplatelet agent other than ASA Bleeding diathesis or coagulation disorderHistory of ICH Gastrointestinal bleeding within 6 months Major surgery within 30 days	Cardioembolic cause i.v or i.a. thrombolysis Mechanical thrombectomy Planned endoarterectomy Planned anticoagulation or specific antiplatelet agent other than ASA Bleeding diathesis or coagulation disorder History of ICH Gastrointestinal bleeding within 6 months Major surgery within 30 days	i.v. thrombolysis Mechanical thrombectomy Planned endovascular revascularization Planned us of anticoagulation or antiplatelet therapy	i.v. thrombolysis Clear indication for anticoagulant or long term antiplatelet drugs Planned surgery or interventional treatment Anticoagulation within 10 days before Gastrointestinal bleeding or major surgery within 3 months Previous ICH
**Mean time from index event to randomization**	within 24 h	within 24 h	within 12 h	within 24 h
**Drugs**	Ticagrelor vs. ASA	Ticagrelor + ASA vs. ASA	Clopidogrel + ASA vs. ASA	Clopidogrel + ASA vs. ASA
**Dosageregimens**	Ticagrelor: 180 mg loading dose, followed by 90 mg twice ASA: 300 loading dose, followed by 100 mg daily	Ticagrelor: 180 mg loading dose, followed by 90 mg twice ASA: 300–325 loading dose, followed by 75–100 mg daily	Clopidogrel: 600 mg loading dose, followed by 75 mg daily ASA: 50 to 325 mg daily	Clopidogrel: 300 mg loading dose, followed by 75 mg daily ASA: clinician-determined dose of 75 to 300 mg on day 1; then 75 mg daily
**Duration of treatment**	90 days	30 days	90 days	21 days clopidogrel + ASA vs. ASA then 69 days clopidogrel alone or ASA alone in respective arm
**Primaryefficacy end point**	Composite of stroke, myocardial infarction, or death within 90 days	Composite of stroke or death within 30 days	Composite of ischemic stroke, myocardial infarction, death from an ischemic vascular event, at 90 days	Ischemic or hemorrhagic stroke at 90 days

**Table 2 jcm-10-01721-t002:** Efficacy and Safety Outcomes.

	SOCRATES [9]	THALES [5]	POINT [10]	CHANCE [11,12]
**Efficacy Outcome**	HR (95% CI)	HR (95% CI)	HR (95% CI)	HR (95% CI)
**Primary end-point**	0.89 (0.78–1.01)	0.83 (0.71–0.96)	0.75 (0.59–0.95)	0.69 (0.58–0.82) ^§^
**Stroke/Death**	/	0.83 (0.71–0.96)	/	/
**Stroke**	0.86 (0.75–0.99)	0.81 (0.69–0.95)	0.74 (0.58–0.94)	0.68 (0.57–0.81)
**Death from any cause**	1.18 (0.83–1.67)	1.33 (0.81–2.19)	1.51 (0.73–3.13)	0.97 (0.40–2.33)
**Cardiovascular death**	1.18 (0.75–1.85)		1.51 (0.43–5.35) *	1.16 (0.35–3.79)
**Ischemic stroke**	0.87 (0.76–1.00)	0.79 (0.68–0.93)	0.72 (0.56–0.92)	0.67 (0.56–0.81)
**Safety Outcome**				
**Major bleeding**	0.83 (0.52–1.34)	3.99 (1.74–9.14)	2.32 (1.10–4.87)	0.94 (0.24–3.79)
**ICH/Fatal bleeding**	/	3.66 (1.48–9.02)	/	/
**ICH**	0.68 (0.33–1.41)	3.33 (1.34–8.28)	1.01 (0.14–7.14)—symptomatic ICH	/
**Haemorragic stroke**	/	10 (0.2%) vs. 2 (<0.1%)	1.68 (0.40–7.03)	1.01 (0.38–2.70)
**Moderate/Severe bleeding**	1.32 (0.99–1.76) ^^^	3.27 (1.67–6.43)	2.45 (1.01–5.90) °	1.41 (0.95–2.10)

^§^ Secondary end point: composite of stroke, myocardial infarction, or death from cardiovascular causes; * Death from ischemic vascular causes; ^ Major or minor bleeding; ° Major hemorrhage other than intracranial hemorrhage.

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
