# Peer review of "Oral Antiplatelet Therapy for Secondary Prevention of Non-Cardioembolic Ischemic Cerebrovascular Events"

_jcm, 2021, doi:10.3390/jcm10081721_

Round 1
Reviewer 1 Report
The authors present a well-structured and in-depth described review on antiplatelet therapy for secondary prevention of non-cardioembolic ischemic stroke. The overview is of major clinical interest due to recent studies adding knowledge to this important topic. The manuscript is well-structured and well-written. The Tables add relevant structured information. I have only minor comments.
Please improve grammar and consider some rephrasing: e.g. S6, line 115: “The association…” instead of “combination”.
Please also comment on the role of intravenous thrombolysis with regard to early DAPT. Or do we face a research gap regarding this topic?
Author Response
The authors present a well-structured and in-depth described review on antiplatelet therapy for secondary prevention of non-cardioembolic ischemic stroke. The overview is of major clinical interest due to recent studies adding knowledge to this important topic. The manuscript is well-structured and well-written. The Tables add relevant structured information. I have only minor comments.
Author reply: we thank the reviewer for the positive comments.
Please improve grammar and consider some rephrasing: e.g. S6, line 115: “The association…” instead of “combination”.
Author reply: we checked the manuscript for grammar errors and typos
Please also comment on the role of intravenous thrombolysis with regard to early DAPT. Or do we face a research gap regarding this topic?
Author reply: We have now specified in the conclusions that ‘Conversely, data on the use of thrombolysis in patients on DAPT (…) are less conclusive. Although current guidelines allow for thrombolysis in patients with stroke under DAPT [7], doubts remain on the safety and the beneficial effects of thrombolytic therapy in this setting, leading to undertreatment’.
Reviewer 2 Report
The introduction is quite short. The authors should include in the introduction refers to the evolution of stroke rates incidence and mortality. It will be also interesting to do a short reference to the main therapeutic resources as well as temporal trends.
Author Response
The introduction is quite short. The authors should include in the introduction refers to the evolution of stroke rates incidence and mortality. It will be also interesting to do a short reference to the main therapeutic resources as well as temporal trends.
Author reply: We thank the reviewer for the suggestion. We have now added in the introduction the following sentence ‘Due to early identification of symptoms, the improvement of primary prevention strategies and in-hospital management, the incidence of stroke and the related short and long-term mortality has been steadily decreasing in recent decades, especially in subjects over 65 and in the most advanced countries.’ and quoted ref 2.
Reviewer 3 Report
Minor revisions:
- Author must provide list of Abbreviations that all are used in manuscript for readers. It is difficult to follow short forms in manuscript if full list of Abbreviations is not available.
In table 2, what is HR as efficacy outcome?
- After the abstract, manuscript lacking Introduction as heading??
otherwise, abstract is to long for paper?
- In Low Dose ASA section, Author state that on page 4, line no 71 about The CAST trial was entirely conducted in China and included 21.106 patients admitted for an acute IS [10].
Author must correct total patients’ numbers, looks like comma is missing in place of full stop.
Author Response
- Author must provide list of Abbreviations that all are used in manuscript for readers. It is difficult to follow short forms in manuscript if full list of Abbreviations is not available.
Author reply: A list of abbreviations has been now provided before the abstract
In table 2, what is HR as efficacy outcome?
Author reply: Hazard ratios have been commonly used to describe the outcome of therapeutic trials where the question is to what extent treatment can shorten the duration of the illness.
- After the abstract, manuscript lacking Introduction as heading??
otherwise, abstract is to long for paper?
Author reply: We have now better specified the difference between abstract and introduction